# Functional Fiber Reduces Mice Obesity by Regulating Intestinal Microbiota

**DOI:** 10.3390/nu14132676

**Published:** 2022-06-28

**Authors:** Mengdi Zhang, Jianhua Liu, Chen Li, Jianwei Gao, Chuanhui Xu, Xiaoyu Wu, Tiesheng Xu, Chenbin Cui, Hongkui Wei, Jian Peng, Rong Zheng

**Affiliations:** 1Department of Animal Genetics and Breeding, College of Animal Science and Technology, Huazhong Agricultural University, Wuhan 430070, China; zhangmd821.edu.cn@webmail.hzau.edu.cn (M.Z.); liujianhua@webmail.hzau.edu.cn (J.L.); lichen14@webmail.hzau.edu.cn (C.L.); jianweig@webmail.hzau.edu.cn (J.G.); pjxutiesheng@163.com (T.X.); 2Department of Animal Nutrition and Feed Science, College of Animal Science and Technology, Huazhong Agricultural University, Wuhan 430070, China; xuchuanhui001@webmail.hzau.edu.cn (C.X.); wuxiaoyu@webmail.hzau.edu.cn (X.W.); cuichenbin@webmail.hzau.edu.cn (C.C.); weihongkui@mail.hzau.edu.cn (H.W.); pengjian@mail.hzau.edu.cn (J.P.); 3The Cooperative Innovation Centre for Sustainable Pig Production, College of Animal Science and Technology, Huazhong Agricultural University, Wuhan 430070, China

**Keywords:** obese mice, metabolic syndrome, intestinal microbes, functional fiber, intestinal barrier

## Abstract

Obesity may cause metabolic syndrome and has become a global public health problem, and dietary fibers (DF) could alleviate obesity and metabolic syndrome by regulating intestinal microbiota. We developed a functional fiber (FF) with a synthetic mixture of polysaccharides, high viscosity, water-binding capacity, swelling capacity, and fermentability. This study aimed to investigate the effect of FF on obesity and to determine its prevention of obesity by modulating the gut microbiota. Physiological, histological, and biochemical parameters, and gut microbiota composition were investigated in the following six groups: control group (Con), high-fat diet group (HFD), low-fat diet group (LFD, conversion of HFD to LFD), high-fat +8% FF group (8% FF), high-fat +12% FF group (12% FF), and high-fat +12% FF + antibiotic group (12% FF + AB). The results demonstrated that 12% FF could promote a reduction in body weight and epididymal adipocyte area, augment insulin sensitivity, and stimulate heat production from brown adipose tissue (BAT) (*p* < 0.05). Compared with the HFD, 12% FF could also significantly improve the intestinal morphological integrity, attenuate systemic inflammation, promote intestinal microbiota homeostasis, and stabilize the production of short-chain fatty acids (SCFAs) (*p* < 0.05). Consistent with the results of 12% FF, the LFD could significantly reduce the body weight and epididymal adipocyte area relative to the HFD (*p* < 0.05), but the LFD and HFD showed no significant difference (*p* > 0.05) in the level of inflammation and SCFAs. Meanwhile, 12% FF supplementation showed an increase (*p* < 0.05) in the abundance of the *Bifidobacterium*, *Lactococcus*, and *Coprococcus* genus in the intestine, which had a negative correlation with obesity and insulin resistance. Additionally, the treatment with antibiotics (12% FF + AB) could inhibit the effect of FF in the HFD. The Kyoto Encyclopedia of Genes and Genomes (KEGG) function prediction revealed that 12% FF could significantly inhibit the cyanogenic amino acid metabolic pathway and decrease the serum succinate concentration relative to the HFD group. The overall results indicate that 12% FF has the potential to reduce obesity through the beneficial regulation of the gut microbiota and metabolites.

## 1. Introduction

Overweight and obesity have become a growing epidemic all over the world, and in 2020, the World Health Organization estimated approximately 1.9 billion overweight and 600 million obese adults worldwide [1]. Obesity can cause metabolic disorders in the body, thus greatly increasing the possibility of type 2 diabetes, cardiovascular disease, nonalcoholic fatty liver, and cancer [2,3,4].

The microbiota in obese individuals play a critical role in the progress of many diseases by influencing nutrient digestion, energy metabolism, glucose metabolism, cholesterol metabolism, and chronic inflammatory progress, leading to enhanced adiposity [5,6,7]. Obese individuals were reported to have a decrease in the abundance of *Bacteroidetes* and an increase in the abundance of *Firmicutes*, coupled with an increase in the relative abundance of conditionally pathogenic bacteria *Atopobium* sp. and *Proteobacteria* in the intestine and a reduction in antimicrobial peptides and secreted mucins, leading to increased intestinal permeability [8,9]. Transplantation of gut microbiota isolated from HFD-induced obese mice into lean germ-free animals could increase body weight and the severity of metabolic syndrome in recipient mice [10]. These reports suggested that gut microbiota may be a target for obesity prevention and treatment.

Dietary fiber (DF), a complex polysaccharide derived from plants, can escape absorption in the small intestine during digestion and modify the microecological environment of the intestine by providing substrates for microbial growth [11,12]. Recent studies have shown the potential of a DF-containing diet to prevent obesity [13]. First, DF’s physicochemical properties (stickiness, fermentability, etc.) had a protective effect against obesity [14], because viscous fibers can prolong gastric emptying and small intestinal transport time, thus converting consumed nutrients into absorbable components [15]. Second, DF can improve energy homeostasis and prevent obesity by boosting the abundance and diversity of obesity-related beneficial gut microbiota [16,17], thus decreasing the ratio of *Firmicutes*/*Bacteroidetes* (F/B ratio) at the phylum level and increasing the relative abundance of *Roseburia* at the genus level [18]. Third, DF can be fermented by gut microbiota to produce short-chain fatty acids (SCFAs) [19], which play important roles in maintaining health, energy metabolism, and preventing certain diseases by lowering the gut luminal pH, inhibiting pathogenic or detrimental gut bacteria, and reducing lipopolysaccharides (LPS) and metabolically detrimental compounds [17,20,21,22]. Meanwhile, the inclusion of fiber in an HFD was also shown to reduce peripheral blood inflammation levels and promote lipolysis through the cyclic adenosine monophosphate (cAMP) pathway, and stimulate brown adipose tissue (BAT) browning by activating uncoupling protein 1 (UCP1) in mice [23,24]. These reports suggested that DF can regulate gut microbiota to prevent obesity.

In our laboratory, we developed a functional fiber (FF) with good viscosity, water solubility, and rapid fermentation. Previous studies revealed that 6% FF could promote the enrichment of the beneficial bacteria *Allobaculum* and *S24-7* family in the intestine, mitigate high-fat-induced intestinal damage, alleviate host inflammation, and improve insulin resistance in HFD-fed obese mice, but could not improve obesity symptoms [25]. The purpose of the present study was to evaluate the functional effects of FF on HFD-fed obese mice by adding 8% or 12% FF to an HFD. Meanwhile, the effects of FF on obesity status, gut microbiota, and metabolites in obese mice were also investigated.

## 2. Materials and Methods

### 2.1. Animal Diets

The experimental dietary formula is shown in Appendix A. The energy content of the low-fat basal diet (LFD) was 3.6 kcal/g, with 19% kilocalories (kcals) from protein, 71% from carbohydrates, and 10% from fat. The calorific value of the high-fat basal diet (HFD) was 5 kcal/g, with 19.4% of the calories from protein, 20.6% from carbohydrate, and 60% from fat. All the feeds were purchased from Trophic Animal Feed High-Tech Corp. Ltd. (Nantong, China). The basal feed was supplemented with a suitable amount of FF (14.3% guar gum (Shangdong Yunzhou Science and Technology Corp., Ltd., Heze, China) and 85.7% pregelatinized waxy maize starch (Hangzhou Puluoxiang Starch Corp., Ltd., Hangzhou, China)) to replace cellulose.

### 2.2. Study Design

A total of 70 specific pathogen-free (SPF) 6-week-old male C57BL/6 mice were purchased from Laboratory Animal Center, Huazhong Agricultural University (Wuhan, China), and housed at 22–24 °C under a 12 h light–dark diurnal cycle, with food and water provided ad libitum. After a 7-day adaptation period, the mice were fed with an LFD or an HFD for 9 weeks, followed by weighing the mice, and defining the obese mice as those with at least 20% weight gain compared to the LFD-fed mice. Next, thirty-five diet-induced obese (DIO) mice were randomly divided into six groups: control group (Con), high-fat diet group (HFD), low-fat diet group (LFD), high-fat +8% FF group (8% FF), high-fat +12% FF group (12% FF), and high-fat +12% FF + antibiotic group (12% FF + AB) (*n* = 7/group), followed by feeding an HFD or an FF-supplemented HFD for 12 weeks. The Con group of lean mice was fed the LFD during the same period (*n* = 7). The experimental dietary formula is shown in Appendix A. The animal experiments were approved by the Animal Care and Use Committee of the Huazhong Agricultural University (Wuhan, China) (HZAUMO-2018-047).

### 2.3. Antibiotic Treatment

The 12% FF + AB group was run in parallel with antibiotics (vancomycin 100 mg/kg, metronidazole 200 mg/kg, neomycin sulfate 200 mg/kg, and ampicillin 200 mg/kg) added to their drinking water, with the water renewed every 3 days for 12 weeks.

### 2.4. Glucose and Insulin Tolerance Tests

An intraperitoneal glucose tolerance test (IGTT) and an intraperitoneal insulin tolerance test (IITT) were conducted at the end of the 11th and 12th weeks. For the IGTT, the mice were fasted overnight, and the baseline blood glucose levels were measured using the Accu-Chek Active Blood Glucose Meter, followed by injecting the mice intraperitoneally with 2 mg glucose/g body weight, and measuring blood glucose levels at 15, 30, 45, 60, and 120 min after the injection. Three days later, the animals were fasted for 6 h, and the fasting glucose levels were determined as described above. For the IITT, these 6 h-fasted mice were injected with 1U insulin/kg body weight (Novolin, Novo Nordisk, Copenhagen, Danish), followed by measuring the blood glucose levels at 15, 30, 45, and 60 min after the injection.

### 2.5. Histopathological Examination

Liver tissue was fixed for 3 days using 4% paraformaldehyde, cut into 4 μm thick sections, and stained with oil red O. The image software Image Pro Plus (version 6.0.0.260, Media Cybernetics Corporation, MD, USA) was used to quantify the red stained sections. White fat from the epididymis, brown fat from the scapula, and liver tissues were fixed in fat fixative for 3 days, and proximal colonic tissue was placed in 4% formalin for 3 days, followed by paraffin embedding, cutting into 4 μm thick sections, and staining with hematoxylin and eosin (H&E). Finally, the tissue morphology was observed under a microscope.

### 2.6. Serum Lipid Profile Analysis

Triglycerides, total cholesterol (TCHO), nonesterified free fatty acids (NEFA), low-density lipoprotein cholesterol (LDL-C), and high-density lipoprotein cholesterol (HDL-C) in the fasting serum (collected at the end of the experiment) were detected using the commercial kit (Nanjing Jiancheng Institute of Biological Engineering, Nanjing, China). Meanwhile, lipopolysaccharide, leptin, tumor necrosis factor, and interleukin-6 concentrations in the mice blood samples were measured using the murine enzyme-linked immunosorbent assay (ELISA) kit (Nanjing Jiancheng Institute of Biological Engineering, Nanjing, China). All test operations were performed according to the kit instructions.

### 2.7. RT-PCR

Total tissue RNA was extracted using TRIZOL reagent (TaKaRa, Tokyo, Japan), followed by purifying the RNA with DNase (Yeasen, Shanghai, China) and using 2 μg of total RNA for reverse transcription. The expression level of related genes was analyzed by quantitative real-time PCR using the specific primers listed in Appendix A, with beta-actin gene being used as the endogenous reference.

### 2.8. Detection of Short-Chain Fatty Acids (SCFAs) in Feces

Briefly, 10–30 mg of the sample was dissolved in 0.5 mL of methanol, followed by homogenization and grinding at 65 Hz for 120 s, and then shaking and mixing for 30 s. After centrifugation at 12,000 r/min and 4 °C for 15 min, 320 μL of the supernatant was transferred to a 1.5 mL EP tube, followed by redissolution in 50 μL of methanol and centrifugation at 12,000 r/min and 4 °C for 15 min. Finally, 100 μL of the supernatant was placed on the machine (GC2010 gas chromatograph, 30.0 m × 0.53 mm CP-Wax 52CB column) to detect the levels of acetic acid, propionic acid, butyric acid, valeric acid, and the branched fatty acids (isobutyric acid and isovaleric acid).

### 2.9. Microbial 16S rRNA Gene Sequencing

Total DNA from fecal samples was extracted by a QIAamp Fast DNA Stool kit (Qiagen, Hilden, Germany), followed by analyzing the DNA quality with 0.8% agarose gel electrophoresis and quantifying the DNA with a UV spectrophotometer. The V3–V4 high variant region of the 16S rRNA gene was amplified with forward primer 341F (5′-ACT CCT ACG GAG GCA GCA GCAG-3′) and reverse primer 806R (5′-GGA CTA CHV GGT WTC TAAT-3′). The PCR products were purified using Ampure XP beads (Beckman, CA USA), followed by using the PCR products to construct libraries and performing the 2 × 250 bp double-end sequencing on an Illumina MiSeq sequencer (Illumina, San Diego, CA, USA) at UW Genetics (BGI, Beijing, China).

### 2.10. The Concentration of Succinic Analysis

The feces and serum were collected to detect the concentration of succinic using the ELISA kit at the end of the animal experiment, according to the manufacturer’s instructions (Shanghai Zeye Biotechnology Co., Ltd., Shanghai, China).

### 2.11. Data Statistical Analysis

All data are presented as mean ± standard error of mean (SEM) (unless otherwise noted). In this study, the Student’s t-test procedure of SAS software (SAS 9.2, SAS Inst., Inc., Cary, NC, USA) was used to evaluate possible differences between the two groups of data at * *p* < 0.05 or ** *p* < 0.01.

## 3. Results

### 3.1. 12% FF Promotes Weight Loss and Improves Obesity Symptoms during HFD Feeding

In this study, a DIO animal model was used to investigate the effects of FF on obesity by feeding the mice with an HFD for 9 weeks and randomly dividing the DIO mice into a control group (Con), high-fat diet group (HFD), low-fat diet group (LFD, conversion of HFD to LFD), high-fat +8% FF group (8% FF), high-fat +12% FF group (12% FF), and high-fat +12% FF + antibiotic group (12% FF + AB) (Appendix A). Figure 1 shows the effects of functional fiber on body weight, adipose tissue, and insulin sensitivity in the obese mice. In Figure 1A–D, it is shown that compared with the HFD group, 8% FF supplementation had no effect on body weight and adipose tissue (*p* > 0.05), in contrast to the significant reduction in body weight and adipose tissue in the LFD group (*p* < 0.01) and the 12% FF group (*p* < 0.05). In Figure 1B, 8% or 12% FF is seen to have no notable influence on energy intake (*p* > 0.05), in contrast to the significant reduction in energy intake in the LFD group (*p* < 0.01). Meanwhile, the treatment of antibiotics was shown to inhibit the effect of 12% FF on decreasing obesity and epididymal fat area (Figure 1A–D). Moreover, compared with the HFD group, 12% FF supplementation could significantly reduce the area under the curve (AUC) in the intraperitoneal glucose tolerance test (IGTT) (*p* < 0.05) (Figure 1E), in contrast to no significant effect on the AUC in the intraperitoneal insulin tolerance test (IITT) (*p* > 0.05) (Figure 1F). In Figure 1E,F, the LFD is seen to significantly reduce the AUC in both the IGTT and the IITT (*p* < 0.01).

The imbalance of lipid metabolism is one of the main causes of lipid accumulation in obese mice. In Figure 1G, the 12% FF and HFD groups are seen to have no significant difference in the levels of total cholesterol (TCHO) and low-density lipoprotein cholesterol (LDL-C) (*p* > 0.05), but 8% or 12% FF could significantly increase serum nonesterified free fatty acids (NEFA) and high-density lipoprotein cholesterol (HDL-C) relative to the HFD (*p* < 0.05), and the effect of FF could be reversed by adding antibiotics. The above results indicated that 12% FF supplementation could improve obesity symptoms, while antibiotic supplementation could inhibit the FF function.

### 3.2. 12% FF Prevents HFD-Induced Adipose Accumulation and Liver Steatosis

Fat accumulation in the liver is an important indicator of HFD-induced metabolic dysregulation. Figure 2 shows the FF effects on the liver in obese mice. In Figure 2A–C, the 8% or 12% FF and HFD groups are seen to have no significant difference in liver weight, triglycerides, and total cholesterol (*p* > 0.05). Oil red O staining of the liver tissue indicated that the 8% FF or 12% FF group had no effect on the size of hepatic fat bubbles relative to the HFD group (*p* > 0.05) (Figure 2D–E).

H&E staining of the liver tissue unveiled that 8% FF or 12% FF supplementation could alleviate hepatic steatosis, which could be significantly reversed by adding antibiotics (12% FF vs. 12% FF + AB) (*p* < 0.05) (Appendix A). Transaminase and bilirubin (BIL) can reflect liver inflammation and liver function. Compared with the HFD, 12% FF could significantly decrease the levels of T-BIL and aspartate aminotransferase (AST) (*p* < 0.05) (Figure 2F–H). Compared with the HFD group, the LFD and 8% or 12% FF groups significantly reduced the mRNA expression of acetyl-CoA carboxylase 1 (*ACC1*), stearoyl-CoA desaturase (*SCD1*), fatty acid synthase (*FAS*), sterol regulatory element-binding protein-1c (*SREBP-1C*), Peroxisome proliferator activated receptor-alpha (*PPAP-α*), and lipid droplet fusion-related gene cell death-inducing DNA fragmentation factor 45-like effector C (*CIDEC*) (*p* < 0.05) (Figure 2I). The above results indicated that 12% FF could enhance liver function, but antibiotic supplementation could reverse the FF-induced reduction in lipid accumulation in obese mice.

### 3.3. 12% FF Reduces Systemic Inflammation in Serum of HFD-Fed Mice by Protecting the Integrity of Gut Epithelium

An important characteristic of obesity is the infiltration and activation of the immune cells in adipose tissues, along with elevated levels of inflammatory factors and inflammatory reactants in peripheral circulating blood. We determined the levels of inflammatory factors in the fasting serum using an enzyme-linked immunosorbent assay (ELISA) method. Figure 3 shows the effects of functional fiber on the inflammatory indexes, gut health, and barrier integrity of obese mice. Compared to the HFD, the 8% or 12% FF group significantly reduced the leptin concentration (*p* < 0.05) (Figure 3A) as well as the levels of lipopolysaccharides (LPS) (*p* < 0.05) (Figure 3B), coupled with a downtrend in interleukin-6 (IL-6) and tumor necrosis factor (TNF-α) concentrations (*p* > 0.05) (Figure 3C,D). However, compared with 12% FF, the treatment of antibiotics could reduce the level of IL-6 (Figure 3C). Additionally, the 8% or 12% FF and HFD groups showed no significant difference in colonic mucosal layer thickness and crypt length (*p* > 0.05) (Figure 3E,F). Moreover, 8% FF supplementation was seen to upregulate the mRNA expression of tight junction gene *Claudin-1* (*p* < 0.05), while 12% supplementation did not, and there was no significant difference in gene *occludin* expression between 8% FF and 12% FF (*p* > 0.05) (Figure 3G). Compared with the HFD, the LFD showed no effect on the foregoing indicators (*p* > 0.05) (Figure 3). These results indicated that compared to the HFD, 8% or 12% FF supplementation could reduce inflammation markers in obese mice, but the LFD could not.

### 3.4. 12% FF Promotes Brown Fat Tissue Thermogenesis

The activation of brown adipose tissue (BAT) thermogenesis is a potential strategy to combat obesity. Figure 4 shows the effects of functional fiber on brown fat tissue thermogenesis in obese mice. H&E staining analysis of scapular BAT revealed that 12% FF supplementation had no significant effect on BAT lipid content (*p* > 0.05), in contrast to a significant effect in the LFD versus the HFD (Figure 4A,B). Meanwhile, RT-PCR analysis showed that 12% FF supplementation could improve the expression of brown fat thermogenic gene uncoupling protein 1 (*UCP1*) and fatty acid transporter protein (*FABP*) (*p* < 0.05) (Figure 4C,D), but had no significant effect on the expression of cluster of differentiation 36 (*CD36*), carnitine palmitoyltransferase 1beta (*CPT-1β*), and deiodinase, iodothyronine, type II (*DIO2*) (*p* > 0.05) (Figure 4F,G). Compared with the HFD, the LFD was seen to significantly increase the expression of the aforementioned five genes (*p* < 0.01) (Figure 4F,G). The above results indicate that 12% FF intervention can stimulate the expression of thermogenic genes in HFD obese mice, but the effect was more pronounced in the LFD, which may be related to the high amount of fiber addition, increasing nonshivering thermogenesis and stimulating white fat browning [26].

### 3.5. 12% FF Modulates the Composition of Intestinal Microbiota

The gut microbiota of the differentially treated mice was analyzed by deep sequencing the 16S rRNA V3–V4 region in colon content and fecal samples. Figure 5 shows the effects of functional fiber on α and β diversity of gut microbiota as well as the effects of antibiotics on the gut microbiota in obese mice. In the colon content, the Chao1 index and Observed species index were significantly increased in the 8% FF or 12% FF group relative to the HFD group (*p* < 0.05) (Figure 5A,B). At the phylum level, the HFD group showed significant abundance of *Firmicutes* when compared to the Con group (*p* < 0.01), while 12% FF supplementation was seen to significantly reduce the abundance of *Firmicutes* and increase the abundance of *Actinobacteria* in the feces (*p* < 0.05) (Figure 5C,D).

Principal Component Analysis (PCA) revealed that microorganisms clustered together in the 8% FF and 12% FF groups and separated from the LFD group (Figure 5E,F). Unweighted Unifrac distance results showed that the Unifrac distance between the control 12% FF group and the HFD group was extremely significantly higher than the distance within the HFD group in the colon (Figure 5G,H) (*p* < 0.01), indicating that the microbial clusters in 12% FF groups were separated from the HFD group.

We explored the alterations of intestinal microorganisms by adding antibiotics coupled with FF supplementation and found that antibiotic-treated mice showed significant reduction (*p* < 0.01) in the index of Chao1, Observed species, Shannon, and Simpson in intestinal flora (Figure 5I). At the phylum and family level, antibiotic treatment increased the abundance of *Aspergillus*, *Enterobacteriaceae*, and *Shigella* (Figure 5J,K). *Enterobacteriaceae* and *Shigella* are mostly conditionally pathogenic bacteria combined with nutrient uptake functions, which may partially explain the reason for antibiotic-induced weight gain in mice.

### 3.6. Correlation between Gut Microbiota and Metabolic Parameters

The colon content and fecal microbiota at the genus level can be clustered into three categories according to basic diet changes and functional fiber supplementation, corresponding to the HFD, 12% FF, and the LFD, respectively (Figure 6A,B). Metastatic analysis revealed an increase in 10 genera and a decrease in 3 genera in the HFD group. Compared with the HFD, the 12% FF group significantly increased in the abundance of *Coprococcus* and unclassified genera (*Ruminococcaceae* and *Coriobacteriaceae*) (*p* < 0.05). In feces, compared with the Con group, the HFD group showed an increase in the abundance of 12 genera and a decrease in the abundance of 4 genera. Meanwhile, the 12% FF group was significantly higher than the HFD group in the relative abundance of *Coriobacteriaceae* and *Bacteria*, but lower in the relative abundance of *Dehalobacterium*, *AF12*, and *Clostridiales* (*p* < 0.05) (Figure 6C,D).

We also analyzed Spearman correlations between colonic/fecal differential bacteria and obesity indicators (12% FF vs. HFD groups). In the colonic content, *Coprococcus* was significantly negatively correlated with insulin resistance and positively correlated with the TBA level, and *oscillospira* was highly significantly negatively correlated with serum leptin levels (*p* < 0.05). *Streptococcus*, *Clostridium Adlercreutzia*, *Lactococcus*, *Subdoligranulum*, and *Leuconostoc* were significantly positively correlated with body weight gain (*p* < 0.05) (Figure 6E). In feces, *Bifidobacterium* was significantly negatively correlated with insulin levels (*p* < 0.05); *Coriobacteriaceae* was significantly negatively correlated with blood glucose levels (*p* < 0.05); *Olsenella* was significantly negatively correlated with body weight gain (*p* < 0.05); *Corynebacterium* was significantly negatively correlated with insulin resistance (*p* < 0.05); *Anaerostipes*, unclassified Bacteria, *Olsenella*, and *Corynebacterium* were all significantly negatively correlated with total bilirubin levels; *Bifidobacterium*, *Anaerostipes*, *Olsenella*, *Corynebacterium*, and unclassified genera from *Bacteria* (*Coriobacteriaceae*) were all significantly negatively correlated with serum leptin levels (*p* < 0.05) (Figure 6F).

The content of SCFAs in feces was determined by gas chromatography–mass spectrometry. The concentration of acetate was significantly higher in the 8% FF and 12% FF groups than in the HFD group (*p* < 0.05), in contrast to no significant difference between the LFD and the HFD (*p* > 0.05) (Figure 6G). Taken together, FF-mediated changes in dominant bacterial genera were closely related to the improvement in host obesity.

### 3.7. Kyoto Encyclopedia of Genes and Genomes (KEGG) Prediction of Succinate Metabolic Pathway

The metabolic pathways with significant differences between groups were identified by the metagenome Seq. KEGG pathway analysis revealed that compared to the HFD group, 12% FF addition could significantly inhibit the cyanogenic amino acid metabolic pathway (*p* < 0.01) (Figure 7A). Meta Cyc functional prediction results showed that 12% FF addition could significantly inhibit the L-arginine, putrescine, and 4-aminobutyric acid degradation pathway, and the L-arginine and L-ornithine degradation pathway (*p* < 0.001) (Figure 7B). It is noteworthy that the end products of both pathways were succinate. Compared to the HFD group, the 8% FF or 12% FF group was lower in fecal and serum succinate concentrations (Figure 7C,D). These results suggest that functional fiber may improve obesity by regulating the metabolism of succinate.

## 4. Discussion

Long-term HFD may promote energy intake and result in the increase in visceral fat deposition, so it is considered as a main factor of obesity [27]. DF is involved in the regulation of energy intake and the metabolism of the body through its fermentative and physicochemical properties [28,29]. In our previous study, 6% FF supplementation was found to have no significant effect on body weight compared to the HFD group [25]. In the present study, 12% FF supplementation under an HFD was shown to significantly reduce body weight (*p* < 0.05), but with no significant effect on food intake (*p* > 0.05). Although FF could slow down the release and absorption of nutrients by reducing the rate of gastric emptying and the intestinal passage of chyme [30], an HFD could stimulate food intake by enhancing the production of lipid-derived endocannabinoid messengers in the gut to limit the effect of functional fiber [31]. Adding guar gum to a high-fat diet was reported to dose-dependently decrease the body weight and adipose weight of mice after 6 weeks of treatment, with a maximal effect at the highest dose of fiber (10% guar gum) [32], inferring that 12% FF addition may reduce body weight in an amount-dependent manner.

We observed a significantly higher serum NEFA and HDL in the 12% FF group compared to the HFD, which is beneficial for health. HDL is usually referred to as good cholesterol because it can promote reverse cholesterol transport, taking up excess cholesterol from peripheral cells and transporting it to the liver for excretion [33]. The increase in NEFA content indicates increased utilization and consumption of neutral fat catabolism [34]. Consistent with the reduced fat mass in the 12% FF group versus the HFD group, a decrease was observed in the circulating levels of leptin, which is produced mostly in adipose tissue and secreted into the blood in proportion to fat depots [35]. Most obese individuals show leptin resistance, characterized by an abnormal increase in serum leptin, while diminishing the effects of leptin on inhibiting appetite, enhancing energy expenditure, and decreasing blood glucose [36]. Long-term DF administration was reported to lower serum leptin level just in obese individuals [37]. Activation of BAT thermogenesis can increase energy expenditure and ameliorate metabolic disease [38]. There is considerable evidence that UCP1 is at the center of BAT thermogenesis and systemic energy homeostasis, and essential to thermogenesis in thermogenic adipocytes [39]. In the present study, the mRNA expression of brown fat thermogenic gene (UCP1) was significantly increased in the 12% FF group relative to the HFD group, which is consistent with previous studies [40]. Collectively, 12% FF addition could promote lipid transport and catabolism and energy expenditure.

It is well known that SCFAs have anti-inflammatory effects through the inhibition of NF-κB activation in the host immune cells, by binding to G-protein-coupled receptors 43 and 41 (GPR43 and GPR41) [41,42]. It is worth noting that the LFD could not improve inflammation levels and SCFAs compared to the HFD, which was probably due to lack of FF in the diet. An HFD can strongly contribute to the development of low-grade systemic inflammation [43]. In our previous study, 6% FF reduced systemic inflammation when the HFD was converted to the LFD [25]. In the present study, the level of IL-6 had a downward trend in the 12% FF group compared to the HFD. However, we found that 12% + AB significantly reduced the level of IL-6 compared to 12% FF. Battson (2017) found that antibiotic treatment successfully abrogated the gut microbiota, accompanied by significant reductions in IL-6 in DIO mice [44]. We also found that microbiota ablation eliminates 12% FF’s ability to reduce epididymal adipose tissue cells, the level of T-BIL, and the expression of lipogenesis key genes in the liver, and promote thermogenesis. Zou (2018) observed that, upon antibiotic cocktail-mediated suppression of the microbiota, inulin’s ability to reduce weight gain and adiposity, and improve glycemic control was not merely eliminated [45]. Han (2021) also reported that oat fiber improved impaired liver function caused by an HFD via promoting gut microbiota-derived SCFAs [46]. We speculated that the effect of FF on reducing epididymal adipose tissue cells, improving liver function and promoting thermogenesis was microbiome dependent.

The interaction between dietary components and gut microbiota composition was reported to have a great impact on the host metabolism [47]. In the present study, the HFD induced an increase in the abundance of Firmicutes, a decrease in the abundance of Bacteroidetes, and an increase in the F/B ratio in the feces of obese mice, which is consistent with previous studies [9,48]. However, 12% FF supplementation could increase the abundance of potentially beneficial bacteria, such as Bifidobacterium, Lactobacillus, and *Coprococcus*, with the abundance of Bifidobacterium and *Coprococcus* being negatively correlated with obesity and insulin resistance. *Bifidobacterium* and *Lactobacillus* are considered as beneficial members of the microbial community due to their ability to effectively promote body weight loss [49,50,51]. Studies have demonstrated that *Bifidobacterium* can improve fasting serum insulin, restore colonic mucus growth, reduce hepatic triacylglycerol accumulation, and alleviate obesity development by increasing acetate levels in HFD mice [44,52,53]. *Lactobacillus* was also shown to have the potential to increase the production of SCFAs [54]. Here, we also found a positive abundance of *Coprococcus* in the colon, contributing to butyrate production and significantly increasing the concentrations of acetate and total SCFAs in the feces. Butyrate could provide energy to the colon, conducive to protect intestinal health, enhance gastrointestinal function, and reduce inflammation levels [55]. Acetate and butyrate could induce intestinal cupped cell differentiation and mucin-related gene expression to regulate mucus production, and secretion to improve the intestinal barrier function [17]. SCFAs were reported to stimulate mitochondrial fatty acid oxidation by activating the UCP2-AMPK-ACC pathway, to prevent and reverse HFD-induced obesity and insulin resistance [56]. In the present study, the concentrations of acetate and total SCFAs were significantly increased in the feces, suggesting that adding 12% FF to an HFD may promote body weight loss by regulating the structure of intestinal microorganisms for the production of SCFAs to prevent obesity.

Moreover, KEGG pathway analysis showed that the improvement of obesity status by FF supplementation may be associated with the metabolism of succinate. Succinate, the intermediate product of the tricarboxylic acid cycle (TCA) or the Krebs cycle, is considered a fuel substrate for mitochondrial oxidative phosphorylation and has pro-inflammatory effects, modulating local stress, tissue damage, and the immune response [57,58]. Obesity is associated with higher circulating levels of succinate [59]. As reported previously, intracellular succinate regulates intestinal gluconeogenesis and thermogenesis, and extracellular succinate is sensed by its cognate receptor SUCNR1 (GPR91) [60,61]. SUCNR1 was expressed on the surface of many immune cells, but most highly expressed on dendritic cells (DCs) and macrophages to enhance the production of pro-inflammatory cytokines, which in turn contribute to inflammation [62]. SUCNR1 can regulate the metabolic response to obesity, with the succinate–SUCNR1 axis serving as a link between metabolic stress and inflammation [62,63]. As intestinal pathogenic bacteria, *Clostridium difficile* and *Shigella* can use succinate for efficient proliferation, leading to increased intestinal mucosal permeability and vascular endothelial cell damage [63].

## 5. Conclusions

It can be concluded that the inclusion of 12% FF in the HFD could effectively alleviate liver fat accumulation, related metabolic syndrome, and promote weight loss in obese mice. Additionally, 12% FF supplementation could induce an increase in the relative abundance of *Bifidobacterium*, *Lactococcus*, and *Coprococcus*. The improvement of obesity and related metabolic syndrome may be related to the abundance decrease in pathogenic bacteria and the increment of acetate in the colon. Our results suggest the great potential of 12% FF as a dietary supplement for obesity management in high-fat diets.

## Figures and Tables

**Figure 1 nutrients-14-02676-f001:**
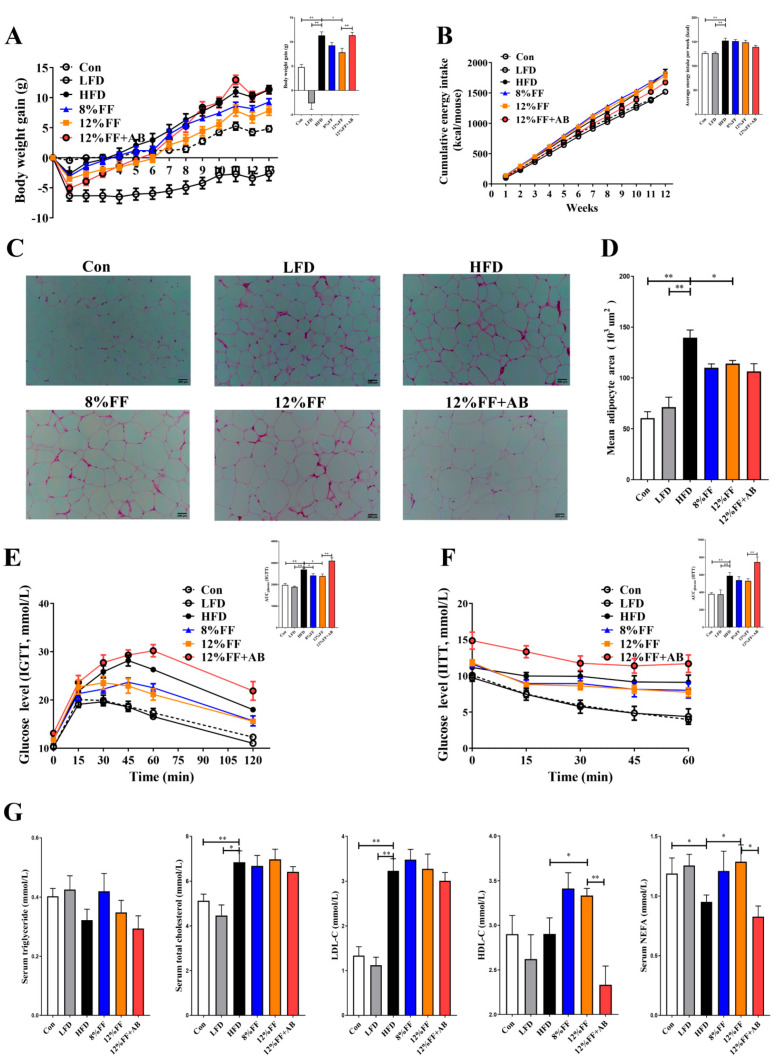
Effects of functional fiber (FF) on body weight, adipose tissue, and insulin sensitivity in diet-induced obese (DIO) mice (*n* = 7). (**A**) Body weight over time and final body weight. (**B**) Accumulative energy intake and average energy intake per week. (**C**) H&E-stained section of epididymal adipose tissue. (**D**) The average area of epididymal adipose tissue cells. (**E**) Intraperitoneal glucose tolerance test (IGTT) and the area under the curve (AUC). (**F**) Intraperitoneal insulin tolerance test and the AUC. (**G**) The concentration of triglycerides, total cholesterol, low-density lipoprotein-cholesterol (LDL-C), high-density lipoprotein-cholesterol (HDL-C), and nonesterified free fatty acids (NEFA) in the serum. The results are shown as the mean ± SEM. Significant differences are expressed as * (*p* < 0.05), and extremely significant differences are expressed as ** (*p* < 0.01).

**Figure 2 nutrients-14-02676-f002:**
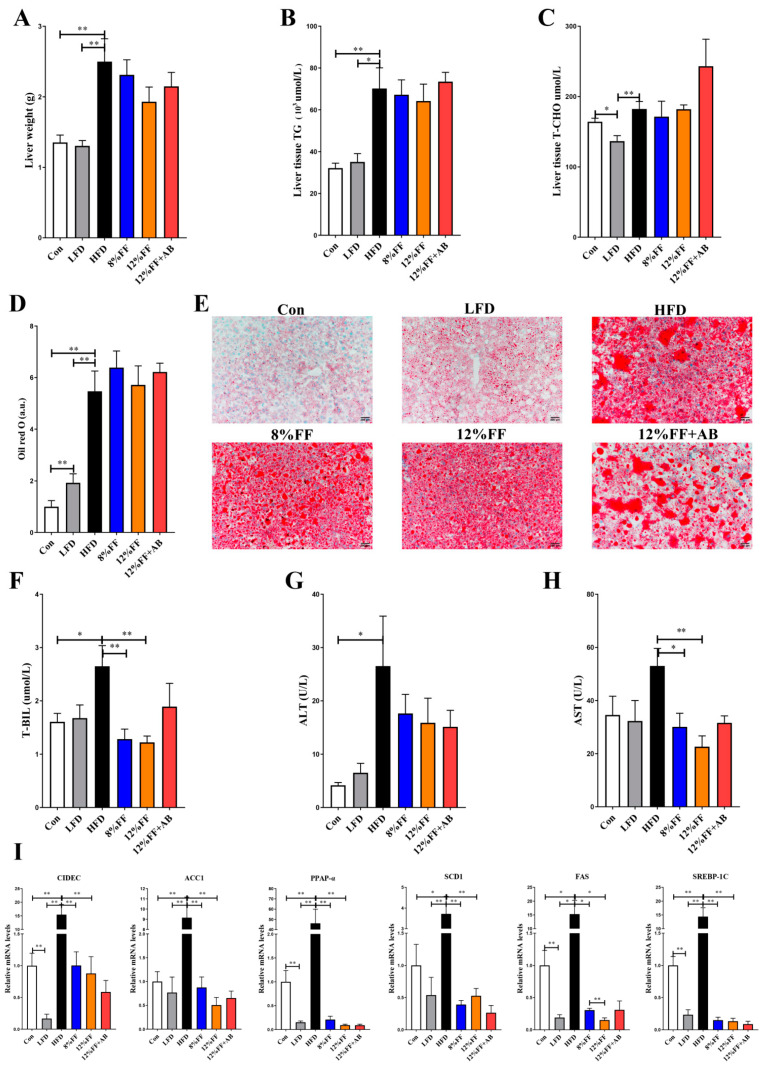
Effects of functional fiber on liver in obese mice (*n* = 7). (**A**) Liver weight. (**B**,**C**) Liver triglycerides and total cholesterol. (**D**,**E**) Liver oil red O staining data. (**F**) Serum total bilirubin (T-BIL). (**G**) Serum alanine aminotransferase (ALT). (**H**) Serum aspartate aminotransferase (AST). (**I**) mRNA expression of liver lipid anabolism genes. The results are shown as the mean ± SEM. Significant differences are expressed as * (*p* < 0.05), and extremely significant differences are expressed as ** (*p* < 0.01).

**Figure 3 nutrients-14-02676-f003:**
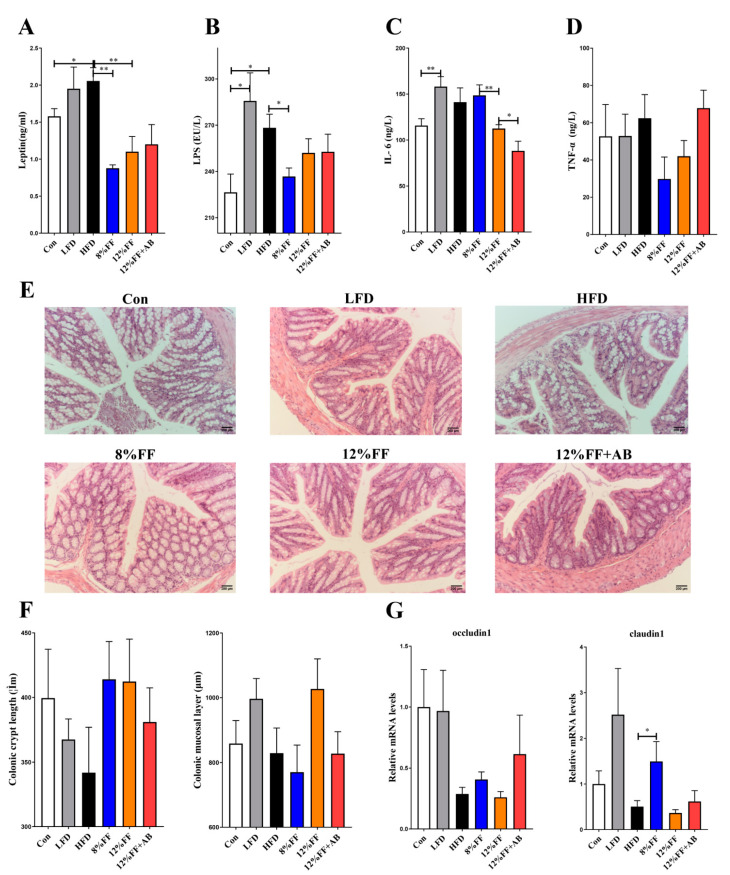
Effects of functional fiber on inflammatory indexes, gut health, and barrier integrity of obese mice (*n* = 7). (**A**) Leptin. (**B**) Lipopolysaccharide (LPS). (**C**) Tumor necrosis factor-α (TNF-α). (**D**) IL-6. (**E**) The H&E-stained images of colon tissue (magnification: 200). (**F**) Depth of colonic crypts and thickness of mucosal layer. (**G**) The mRNA expression of colonic tight junction protein gene *occludin 1* and *Claudin-1*. The results are shown as the mean ± SEM. Significant differences are expressed as * (*p* < 0.05), and extremely significant differences are expressed as ** (*p* < 0.01).

**Figure 4 nutrients-14-02676-f004:**
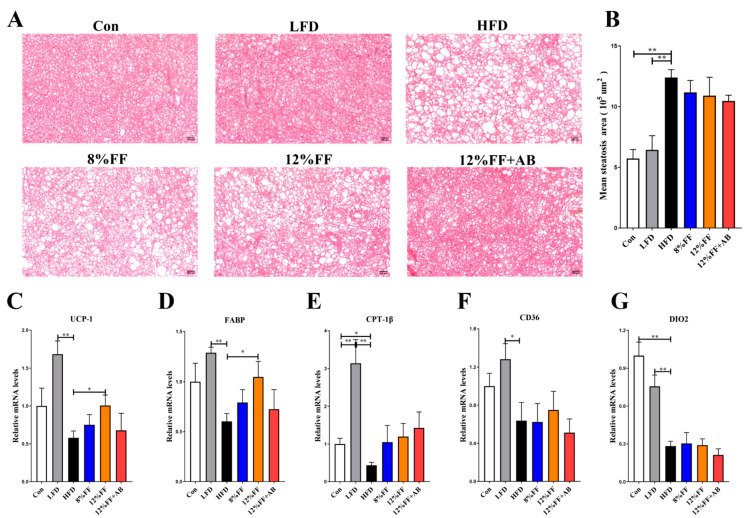
Effects of functional fiber on brown fat tissue thermogenesis in obese mice (*n* = 7). (**A**) H&E staining analysis of brown adipose tissue of scapula (magnification: 200). (**B**) Lipid content in brown tissue of scapula. (**C**–**G**) mRNA expression of thermogenic genes involved in brown adipose tissue of scapula. The results are shown as the mean ± SEM. significant differences are expressed as * (*p* < 0.05), and extremely significant differences are expressed as ** (*p* < 0.01).

**Figure 5 nutrients-14-02676-f005:**
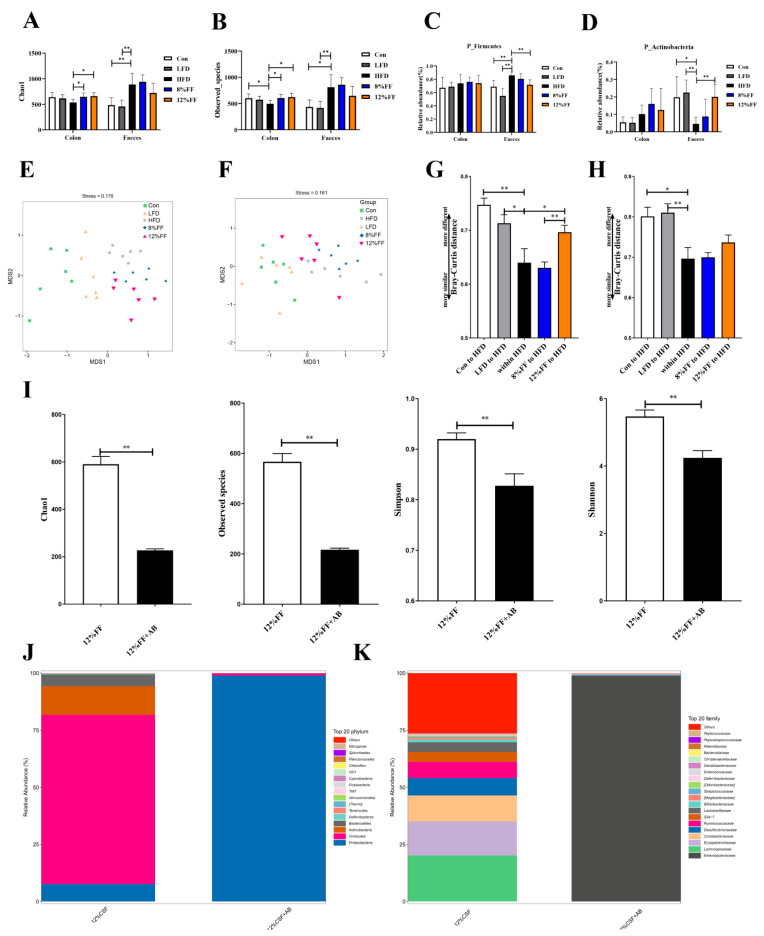
Effects of functional fiber on α and β diversity of gut microbiota, and effects of antibiotics on the gut microbiota in obese mice. (**A**,**B**) Chaol and Observed species index (*n* = 6). (**C**,**D**) The abundance of *Firmicutes* and *Actinobacteria* (*n* = 6). (**E**,**F**) The principal component analysis (PCA) of colon and fecal samples (*n* = 6). (**G**) Bray–Curtis distance between colonic microbiomes (*n* = 6). (**H**) Bray–Curtis distance between fecal microbiomes (*n* = 6). (**I**) Chao1, Observed species, Simpson, and Shannon index (*n* = 7). (**J**) Histogram of colonic microbiota level composition (*n* = 7). (**K**) Histogram of colonic microbial genus level composition (*n* = 7). The results are shown as the mean ± SEM. Significant differences are expressed as * (*p* < 0.05), and extremely significant differences are expressed as ** (*p* < 0.01).

**Figure 6 nutrients-14-02676-f006:**
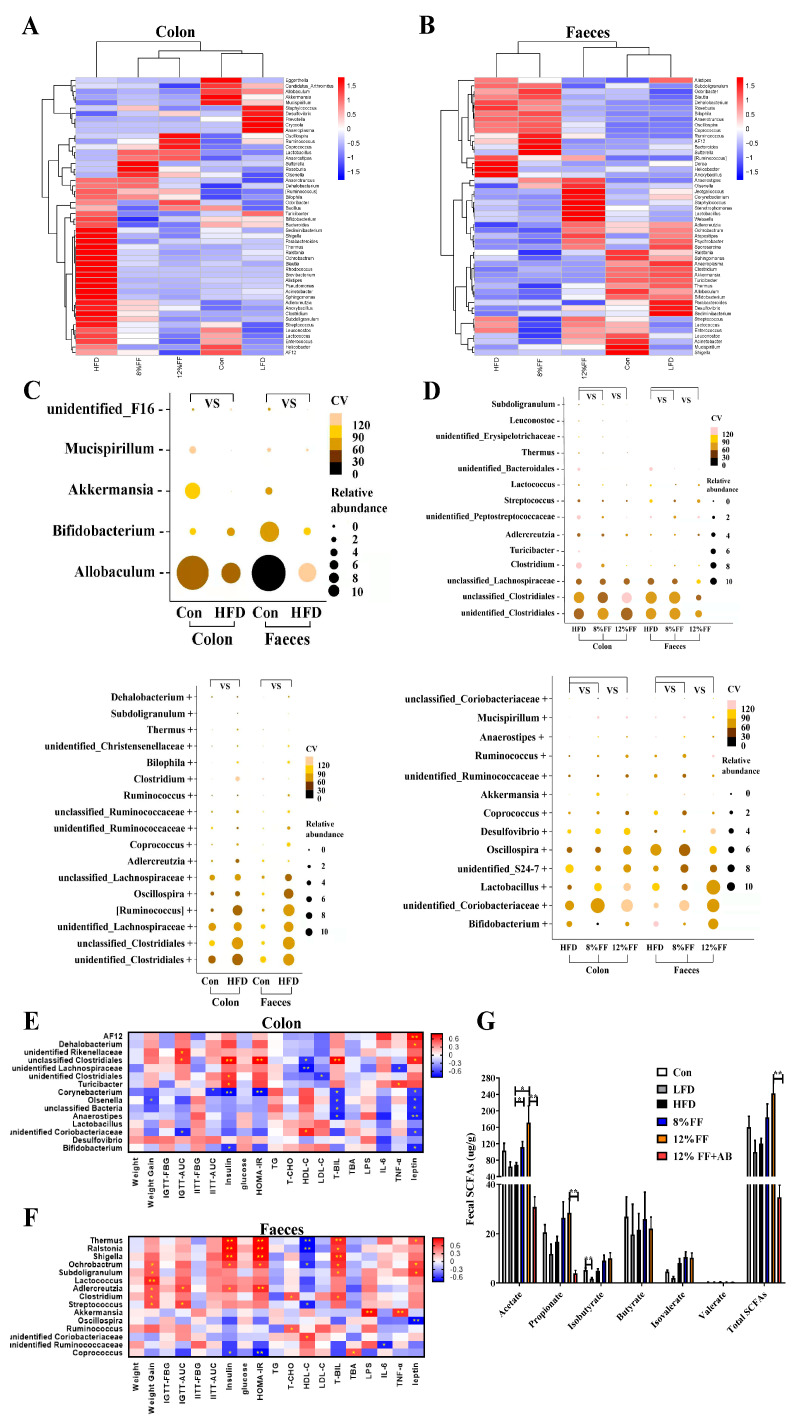
Effects of functional fiber on the genus of fecal microbiota in obese mice (*n* = 6). (**A**,**B**) The heat maps of genus composition of colonic microbial and fecal microbial species clusters. (**C**,**D**) Bacteria with abundance decrease (top) and increase (below) in HFD group vs. Con group and 12% FF group vs. HFD group, respectively. (**E**) The Spearman correlation between colonic differential bacteria and obesity indicators. (**F**) The Spearman correlation between fecal differential bacteria and obesity indicators. (**G**) The effect of functional fiber on the production of SCFAs in obese mice (*n* = 7). The results are shown as the mean ± SEM. Significant differences are expressed as * (*p* < 0.05), and extremely significant differences are expressed as ** (*p* < 0.01).

**Figure 7 nutrients-14-02676-f007:**
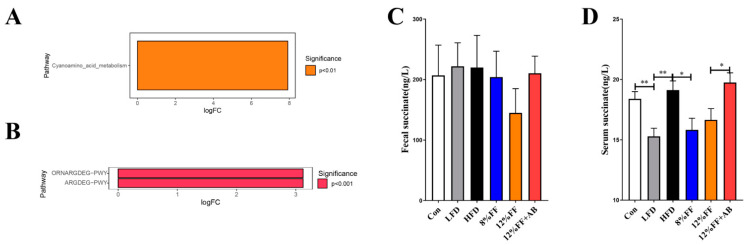
Effects of functional fiber on the metabolic function of gut microbiota in obese mice. (**A**) KEGG function prediction result. (**B**) MetaCyc function prediction result. (**C**) Succinic acid concentration in feces (*n* = 7). (**D**) Succinic acid concentration in serum (*n* = 7). The results are shown as the mean ± standard error (*n* = 7). Significant differences are expressed as * (*p* < 0.05), and extremely significant differences are expressed as ** (*p* < 0.01).

## Data Availability

Not applicable.

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
