# Peer review of "Functional Fiber Reduces Mice Obesity by Regulating Intestinal Microbiota"

_nutrients, 2022, doi:10.3390/nu14132676_

Round 1

Reviewer 1 Report

Thank you for the opportunity to review this interesting manuscript. I have a few questions and comments which need to be addressed to improve the quality of the manuscript.

1. Are the HFD, HFD with 8% and HFD with 12% isocaloric per gram of feed? Is the FF considered absorbable calories or nonabsorbable. If non-absorbable, how was this accounted in the Kcal/g calculation. Please include g/day as well as kcal/day.  I believe bomb calorimetry for calculation of fecal Kcal loss would be very useful to understand the discrepancy between body weight change and cumulative energy intake. Additionally, calorimetry would be exceptionally useful here to understand where the energy deficit is to produce weight loss with FF, especially since FF plus AB reduced cumulative energy intake but significantly increased body weight. The difference in effect on body weight vs adipocyte mass needs to be explained with FF and FF+AB. 

2. The discussion is lacking a clear explanation of what effects are believed to be due to changes in the microbiome related to FF administration, and what effects are microbiome independent (t bili, liver lipid anabolic genes, IL6, thermogenesis, adipocyte mass).

3. How can steatosis improve which is FF and microbiome dependent, but not liver weight, liver TG, or liver oil Red 0?

Minor:

1. Methods- Was the antibiotic treatment for the duration of the 12% FF trial? Please describe more clearly in the methods. 

2. Figure S1- Remove the term "slaughter" sampling. no yellow dot is noted on the diaphragm. If this refers to euthanasia, please label appropriately. GLP-1 and PYY are on this diagram but not in the methods or manuscript.

3. Table S1- What is "Feed calorie situation?" Below this Row, the fat percent for LFD and HFD are both listed as 60.

4. For the AST figure, there is no 12%FF+AB displayed

5. Do not include p values for those that are not significant on graphs.

6. The trend lines between groups Fig 5A-D does not make sense. Please display as columns. While Figure 5I makes sense, it seems odd to not have 12FF +AB with figures 5A-D as done through out the rest of the manuscript.

Reviewer 2 Report

In this study, Zhang and colleagues investigated the role of dietary fiber in reducing obesity in mice and explored the role of the gut microbiota.
The topic of the paper is very interesting and may help to clarify the role of a much discussed dietary component such as dietary fiber in a pathogenic context, as the role of dietary fiber is very important as it is recommended in most slimming diets.
The manuscript is well written, the experimental approach is correct, and the results are interesting.
One should only ask the authors to better describe the experimental design:
How many mice were used? With 5 experimental groups of 5 mice each, the final number should be 35 and not 28 as stated.
Also, be more specific about the LFD administered to the mice in the first paragraph of the results section.

I suggest moving Figure S1 to the main text, furthermore, the FF 12% plus antibiotic group is lacking
